# PROPERTY INFERENCE ATTACKS AGAINST $t$-SNE PLOTS

## ABSTRACT

With the prevailing of machine learning (ML), researchers have shown that ML models are also vulnerable to various privacy and security attacks. As one of the representative attacks, the property inference attack aims to infer the private/sensitive properties of the training data (e.g., race distribution) given the output of ML models. In this paper, we present a new side channel for property inference attacks, i.e., $t$-SNE plots, which are widely used to show feature distribution or demonstrate model performance. We show for the first time that the private/sensitive properties of the data that are used to generate the plot can be successfully predicted. Briefly, we leverage the publicly available model as the shadow model to generate $t$-SNE plots with different properties. We use those plots to train an attack model, which is a simple image classifier, to infer the specific property of a given $t$-SNE plot. Extensive evaluation on four datasets shows that our proposed attack can effectively infer the undisclosed property of the data presented in the $t$-SNE plots, even when the shadow model is different from the target model used to generate the $t$-SNE plots. We also reveal that the attacks are robust in various scenarios, such as constructing the attack with fewer $t$-SNE plots/different density settings and attacking $t$-SNE plots generated by fine-tuned target models. The simplicity of our attack method indicates that the potential risk of leaking sensitive properties in $t$-SNE plots is largely underestimated. As possible defenses, we observe that adding noise to the image embeddings or $t$-SNE coordinates effectively mitigates attacks but can be bypassed by adaptive attacks, which prompts the need for more effective defenses.

## 1 INTRODUCTION

Machine learning (ML) models are becoming powerful and can be used as a feature extractor to generate representations (also known as embeddings) for the input data (He et al., 2016; Sandler et al., 2018; Huang et al., 2017). However, such representations are still in high-dimensional space (e.g., 512 dimensions for the ResNet-18). To better understand the representations of data or demonstrate the model's performance, people usually use dimension reduction techniques such as $t$-SNE (van der Maaten & Hinton, 2008) (abbreviation for t-distributed stochastic neighbor embedding) to reduce high-dimensional representations into a 2-dimensional space for visualization.

Despite being powerful, ML models are also shown to be vulnerable to various privacy attacks that aim to reveal the sensitive information of the training dataset given access to the target model. *Property inference attack* (Ganju et al., 2018; Zhou et al., 2022; Mahloujifar et al., 2022) is one of the representative attacks whereby the adversary aims to infer the sensitive global properties of the training dataset (e.g., the race distribution) from the representations generated from the target model.

$t$-SNE plots, on the other hand, are condensations of data representations. Such plots are usually considered to be safe and shared with the public via scientific papers or blogs. However, it is unclear whether such plots would leak sensitive property information about the data as well.

**Our Work.** In this paper, we take the first step toward understanding the privacy leakage from $t$-SNE plots through the lens of the property inference attack against such plots. Here, we consider the general property as the macro-level information of the dataset, e.g., race distributions. Note that this property is not necessarily related to the $t$-SNE plots, for instance, the $t$-SNE plot is used to show how distinguishable the gender distribution is, and the adversary can infer the race distribution of

the data used to generate the plot. A successful attack may cause severe consequences as it provides additional information to the adversary, which is often sensitive and should be protected. Also, it can be used to audit the fairness of the dataset (Buolamwini & Gebru, 2018).

In this work, we first systematize the threat model and attack methodology, which is straightforward. We assume that the adversary can access the victim's $t$-SNE plot and may have knowledge of the distribution of the target dataset, but they do not necessarily have access to the target model. To infer the general property of the samples in the $t$-SNE plot, the adversary first samples groups of images from a shadow dataset with different properties (e.g. different proportions of males). Then, those groups of images will be used to query the shadow model to get representations and generate groups of $t$-SNE plots. An attack model is trained based on the <plot, property> pairs, which is an image classifier that can distinguish between $t$-SNE plots with different labels. Once well trained, the attack model can then be used to infer the property of public $t$-SNE plots.

Our evaluations on both classification and regression tasks show that the proposed attack can achieve high accuracy, on some of the datasets like CelebA and LFW. For instance, the accuracy for predicting the proportion of males on CelebA $t$-SNE plots reaches 0.92, and the average regression error of predicting precise proportions is 0.02. We also study the reason for the relatively poor attack performance on the other datasets (e.g. FairFace) or attributes (e.g. Oval Face). We discover that this is due to the less distinguishable representations generated by the target model over these datasets/attributes. Also, we observe that the attack is still effective even when the shadow model is different from the target model. We further demonstrate that our attack is robust with fewer training plots and transferable to different $t$-SNE density settings. For instance, the regression attack model trained on $t$-SNE plots with 750 sample points can reach a low error of 0.04 on $t$-SNE plots with $1,000$ or $500$ sample points. We additionally show that, by mixing only a small number of $t$-SNE plots from a new dataset, our attack can transfer to the new dataset. and we reveal the validity of our attack when the target and shadow models are fine-tuned, which is common in practice (see Section 5.5 for details).

To mitigate the attack, we perturb the image embeddings/$t$-SNE coordinates and discover that it indeed reduces the attack performance to a large extent, e.g., by adding Gaussian noise to the $t$-SNE coordinates, the inference error of regression attack model increases significantly from 0.02 to 0.48. However, such defenses can still be bypassed by an adaptive attacker (see also Section 5.4). In short, our work demonstrates that the published $t$-SNE plots can be a valid side-channel for the property inference attacks. We appeal to our community's attention to the privacy protection of publishing $t$-SNE plots.

## 2 BACKGROUND AND RELATED WORK

$t$-**SNE Plots.** Generally, $t$-SNE (van der Maaten & Hinton, 2008) is used to transform high-dimensional data to low-dimensional (e.g. 2D) data while preserving their relationship, i.e., similar data are mapped to the near space and dissimilar data are projected more distant. To analyze how separable some specific characteristics of the data are, one common practice is to leverage the pre-trained ML models as feature extractors to generate representations for the data and use $t$-SNE to reduce the high dimensional representations into 2D space for visualization, i.e., with $t$-SNE plots (see Figure 4(b) as an example). Also, $t$-SNE plots can be used to show the performance of fine-tuned ML models (see Figure 13 as an example).

**Property Inference Attack.** Property inference intends to gain insights into the global properties of training datasets, which are unconsciously leaked. This poses a threat to the intellectual property of the data owner. In addition, property inference can audit the fairness of datasets, e.g. the gender fairness in benchmark datasets (Buolamwini & Gebru, 2018). Previous work has shown the vulnerability of property inference against machine learning models, including discriminative models and generative models. Ateniese et al. (2015) proposes inference attack against shallow machine learning models, e.g. Hidden Markov Models and Support Vector Machines. Ganju et al. (2018) proposes the first property inference attack against fully connected neural networks. Both of the work assumes that the adversary has white-box access to the machine learning models. Concretely, the adversary first trains shadow models on datasets with different properties, then leverages the parameters of shadow models to train the meta classifier to identify the property of the training dataset. Mahloujifar et al. (2022) conducts the property inference attack by injecting poisonous data into

the training dataset and then querying the target model in a black-box fashion. Zhou et al. (2022) reveals the first set of training property attacks against GANs, where the adversary also has only black-box access to the target model. These previous works focus on the training data of machine learning models, and many assume that the adversary has white-box access to the model. Our work concentrates on both the white-box and black-box property inference attack against $t$-SNE plots, the safety and privacy of which are paid less attention to when published. Also, our proposed attack trains a regression model to precisely infer the property.

Note that in this paper, we consider the adversary is interested in the property of **the data points that are used to generate the $t$-SNE plots**.

## 3 THREAT MODEL AND ATTACK METHODOLOGY

### 3.1 THREAT MODEL

$t$-SNE plots are usually leveraged to show the feature distribution of the data or demonstrate the representation ability of a machine learning model. Such plots are commonly published on websites or in scientific papers. In this work, we propose the first property inference attack against published $t$-SNE plots where anyone observing the plots could be the attacker. The $t$-SNE plot may only present insensitive information, but other sensitive information of the dataset could also be inferred. For instance, a model owner publishes a $t$-SNE plot of human face images, labeled by age, which is intended to show the effectiveness of the age classification ML model. However, the attacker can infer the proportion of males in the samples that are used to draw the $t$-SNE.

**Adversary's Goal.** An adversary aims to infer the sensitive property $\mathcal{P}_{target}$ of the data samples that are used to generate a $t$-SNE plot. And the property is not directly shown by the $t$-SNE plot. To be specific, the property we consider in this paper is the distribution of a certain attribute, e.g. the proportion of males.

**Adversary's Background Knowledge.** We make the following assumptions about the adversary's background knowledge. First, the adversary has access to the $t$-SNE plot and can make adjustments to the plots, e.g. removing axes, labels, plot titles, and changing the color. Second, the adversary is aware of the dataset distribution but does not know which subset of images are used to generate the $t$-SNE plots. This means that the adversary can obtain a shadow dataset and generate $t$-SNE plots with subsets of images that have different property values, e.g., different proportions of males. We later show that even if the shadow dataset comes from a different distribution, with only a few numbers of $t$-SNE plots that share the same data distribution as the target $t$-SNE plot, our attack can successfully infer the property of the data used to generate the $t$-SNE plots. Third, the adversary has a shadow model to embed the images. We first assume the shadow model is the same as the target model. We later show that the attacks are still effective when the shadow model is different from the target model (Figure 2 and Table 4).

### 3.2 ATTACK METHODOLOGY

Briefly, the attack consists of two parts: $t$-SNE generation and attack model training.

$t$-**SNE Generation.** To correctly infer the target property $\mathcal{P}_{target}$ from the given $t$-SNE plot, one possible solution is to construct a set of $t$-SNE plots that have different values of the target property, e.g., different proportions of males in the plot data. Concretely, the adversary first samples many subsets of data from a shadow dataset, each with a different value of the target property. Then, they use each subset of images to query the shadow model and get the image embeddings (output from the second-to-the-last layer of the shadow model). The adversary then projects the embeddings into 2D space and generates a $t$-SNE plot for each subset. The $t$-SNE plots are labeled by the property value of the image subset. Note that if the adversary has white-box access to the target model, they can directly apply the target model as the shadow model to generate $t$-SNE plots.

**Attack Model Training.** With the generated $t$-SNE plots, the adversary trains an ML model to predict the property values of the plots. Note that in this paper, we consider both classification and regression tasks, while previous property inference attacks (Ateniese et al., 2015; Ganju et al., 2018; Mahloujifar et al., 2022) only consider the classification task. The input is the generated $t$-SNE

plots, and the output is the property value of the $t$-SNE plots. Once the attack model is trained, it can be used to infer the properties of public $t$-SNE plots.

## 4 EXPERIMENTAL SETUP

**Datasets and Models.** We consider 4 face datasets in our evaluation, including CelebA (Liu et al., 2015), LFW (Huang et al., 2008), UTKFace (Zhang et al., 2017), and FairFace (Kärkkäinen & Joo, 2021). Regarding the model architectures, we use the ResNet18 (He et al., 2016), ResNet34 (He et al., 2016) and the small version of MobileNetV3 (Howard et al., 2019) pre-trained on ImageNet1k as the target and shadow models' architectures. Also, we use the pre-trained DenseNet121 (Huang et al., 2017) as the attack model's architecture. More details of the datasets and models are in Section A.1 and Section A.2. Each dataset is first randomly divided into two equal parts, i.e., the shadow dataset and the target dataset, without overlapping. In our experiment, the shadow dataset is used to generate $t$-SNEs for training the attack model, while the target dataset is used to generate $t$-SNEs with different property values for the testing.

**Target/Shadow Model and Two Experiment Settings.** To show the effectiveness of the attack, we conduct experiments on both multi-class classification and regression tasks. For both of the tasks, we consider two settings. Setting 1 is called *same shadow model* where the adversary has white-box access to the target model and can directly leverage the target model as the shadow model. Setting 2 is called *different shadow model* where the adversary has to construct the shadow model themselves. By default, we set the shadow model as pre-trained ResNet18. It means that, for setting 1, the target model is pre-trained ResNet18 as well. For setting 2, the target model is pre-trained ResNet34 or MobileNetV3.

$t$**-SNE Generation.** To generate a $t$-SNE plot, we first sample 1,000 images with a specific proportion of a certain attribute from the shadow (target) dataset. Specifically, we query the shadow (target) model with these sampled images and take the output of the second-to-the-last layer as the samples' embeddings to generate the $t$-SNEs. Since we assume the adversary can adjust the color of the target model's $t$-SNE (e.g. using image editing tools), we conduct our attack in a more uniform (and more difficult) setting with no label and color information in $t$-SNEs (see Figure 4(a) as an example of the $t$-SNE plots we use). And the figure size of 300 pixels * 300 pixels.

For multi-class classification, we set five proportions as labels to infer, i.e. $\mathcal{P}_{target} \in \{0.3, 0.4, 0.5, 0.6, 0.7\}$ following (Zhou et al., 2022). For each label, we generate 150 $t$-SNE plots using the shadow model for attack model training, and 50 $t$-SNE plots using the target model for testing. For regression tasks, the proportions range from 0.0 to 1.0, with a stride of 0.01. The attack model's training set contains 40 $t$-SNEs for each label (4,040 in total), while the testing set consists of 10 $t$-SNEs for each label (1,010 in total).

To investigate our attack's transferability, we also evaluate the case where the target $t$-SNEs are generated from the unseen dataset (see Table 1 and Table 2).

**Attack Model Training.** By default, we use the ImageNet pre-trained DenseNet121 with the Adam optimizer as the attack model. For classification, we set the training batch size to 64, the learning rate to 0.0001, the cross-entropy as the loss function, and the training epoch to 30. For regression, we set the training batch size to 128, the learning rate to 0.0001, the MSE as the loss function, and the training epoch to 100.

## 5 EVALUATION RESULTS

### 5.1 RESULTS OF MULTI-CLASS CLASSIFICATION TASKS

We first consider setting 1 where the adversary has white-box access to the target model. We conduct experiments on different datasets, different target properties, and different model architectures.

**Different Datasets.** From Figure 1(a), we observe that our attack can successfully reveal the proportion of males in some of the datasets. For instance, in CelebA, our attack reaches 0.92 accuracy while the baseline accuracy is only 0.20. Another finding is that the inference accuracy varies across different datasets, e.g., 0.92, 0.56, 0.41, and 0.22 on CelebA, LFW, UTKFace, and FairFace, re-

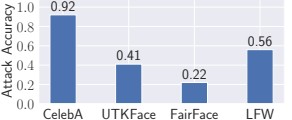 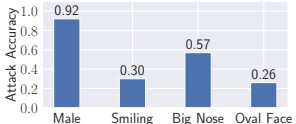 

(a) Accuracy on male proportion of different datasets.
(b) Accuracy on different target properties of CelebA.
(c) Accuracy on male proportion of CelebA with different models.

Figure 1: Multi-classification results under setting 1 for different datasets, target properties and models. The testing data is from the same distribution as the training data. For (a) and (b), both the target model and shadow model are pre-trained ResNet18.

Table 1: Multi-classification transfer study under setting 1. The accuracy values are in bold for the testing data that is from the same distribution as the training data.

| Training Data | Target Property | Cel. | UTK. | Fair. | LFW |
|---|---|---|---|---|---|
| CelebA | Male | **0.92** | 0.28 | 0.21 | 0.29 |
| UTKFace | Male | 0.24 | **0.41** | 0.19 | 0.22 |
| FairFace | Male | 0.20 | 0.22 | **0.22** | 0.21 |
| LFW | Male | 0.11 | 0.21 | 0.20 | **0.56** |

Table 2: Regression transfer study under setting 1. The error values are in bold for the testing data that is from the same distribution as the training data.

| Training Data | Target Property | Cel. | UTK. | Fair. | LFW |
|---|---|---|---|---|---|
| CelebA | Male | **0.02** | 0.38 | 0.35 | 0.27 |
| UTKFace | Male | 0.32 | **0.08** | 0.36 | 0.38 |
| Fairface | Male | 0.21 | 0.22 | **0.17** | 0.24 |
| LFW | Male | 0.32 | 0.37 | 0.41 | **0.07** |

spectively. We credit this to the distinguishability of overall image embedding distribution when the proportion of males changes. To confirm this, for each $t$-SNE plot, we calculate the average embedding of all image embeddings that are used to generate the plot. Then we use $t$-SNE to project these average embeddings of all 750 $t$-SNEs (150 per class) and color them by the property, i.e. proportion of males, to show the distinguishability of different classes. Figure 5 exhibits the results. Figure 5(a) shows five clear dense clusters, indicating that the five classes of $t$-SNEs from CelebA are well distinguishable. The $t$-SNEs of different classes from UTKFace (Figure 5(b)) and LFW (Figure 5(d)) are also clustering well, but slightly sparser than CelebA for each cluster. However, in stark contrast to the other three datasets, the average embeddings of $t$-SNEs of different classes from FairFace are evenly distributed across the plot, in Figure 5(c), indicating that they are difficult to distinguish.

**Different Target Properties.** We further conduct the experiments on different target properties of CelebA, shown in Figure 1(b). We observe that the attack performs well at inferring the proportions of Male (0.92) and Big Nose (0.57) but worse at inferring the proportions of Smiling (0.30) and Oval Face (0.26) in CelebA. We impute this to the bad performance of the feature extractor when distinguishing Smiling faces and Oval Face faces. To verify this, with a pre-trained ResNet18 as the target model, for each attribute, we randomly select 1,000 face images with half of the images having the attribute (e.g., 500 male and 500 female images) and query the target model to obtain the embeddings and generate a $t$-SNE. We mark the two groups with different colors, e.g. half of the images are male, colored by yellow while the other half are female, colored by purple. The results are shown in Figure 6. From Figure 6(a) and Figure 6(b), the two clusters are more discernible than Figure 6(c) and Figure 6(d). This suggests that the pre-trained ResNet18 cannot separate Smiling (Oval Face) faces from not Smiling (Oval Face) faces well, which further makes the $t$-SNE plots with different proportions of Smiling (Oval Face) faces indistinguishable.

**Different Models.** We also find that our attack is effective across different target model architectures. As shown in Figure 1(c), the attack accuracy are 0.92, 0.91, and 0.82, on ResNet18, ResNet34, and MobileNetV3, respectively, which is significantly higher than the baseline accuracy (0.2). Also, MobileNetV3 has lower inference performance compared to ResNets. We suspect the reason is that MobileNetV3 is smaller and has a lower representation ability than ResNets and thus makes the $t$-SNEs less separable.

**Transfer Study.** In a more difficult scenario when the target $t$-SNEs are generated from different datasets (see Table 1), we observe that the attack fails with an accuracy of around 0.2 (random guess baseline). This is expected as the difference across datasets may result in huge changes to

Table 3: Multi-classification results under setting 2. The shadow model is pre-trained ResNet18.

| Target Model | Training Data | Target Property | Test Acc. |
| --- | --- | --- | --- |
| ResNet34 | CelebA | Male | 0.86 |
| MobileNetV3 | CelebA | Male | 0.46 |

Table 4: Regression results under setting 2. The shadow model is pre-trained ResNet18.

| Target Model | Training Data | Target Property | Test Acc. |
| --- | --- | --- | --- |
| ResNet34 | CelebA | Male | 0.03 |
| MobileNetV3 | CelebA | Male | 0.13 |

the $t$-SNEs, making the inference more difficult. We later show in Section 5.5 that the attack can transfer to $t$-SNEs generated from other datasets by introducing a small number of $t$-SNEs from those datasets to further train the attack model (similar to few-shot learning).

**Different Shadow Model.** For setting 2, the shadow model architecture is different from the target model. We use the pre-trained ResNet34 and the small version of MobileNetV3 as the target model, and the pre-trained ResNet18 as the shadow model to infer the proportion of males on CelebA. Results are exhibited in Table 3. Overall, the inference accuracy decreases, compared to setting 1. For instance, the accuracy on male proportion of CelebA is 0.86 under setting 2 with ResNet34 the target model, which, under setting 1, is 0.92. However, the accuracy is still much better than random guessing (0.20). When the target model is MobileNetV3, the inference accuracy severely drops to 0.46. This is because ResNet34 shares similar modules with ResNet18, and thus the image embeddings extracted by ResNet34 (target model), compared to MobileNetV3, are more similar to ResNet18 (shadow model). This will lead to similar $t$-SNE patterns and higher inference accuracy.

To conclude, our results show that the same attribute of different datasets does not equally suffer from the attack, and proportions of some attributes such as Male and Big Nose are more easily inferred than other attributes. We visualize the $t$-SNEs with different proportions of males using Grad-CAM (Selvaraju et al., 2017) in the appendix (see Section A.3). Also, when the shadow model is different from the target model, the attack is more effective when the shadow model shares similar architectures with the target model (e.g., ResNet18 vs. ResNet34).

## 5.2 RESULTS OF REGRESSION TASKS

To generalize the attack and infer the property more precisely, we frame our attack as regression tasks on different datasets, target properties, and model architectures. The evaluation metric is the average error between the predicted proportion and the ground truth value in the regression setting. For instance, if the predicted proportion is $0.10$ and the ground truth is $0.15$, the error for this $t$-SNE plot is $|0.15 - 0.10| = 0.05$. If there are $n$ $t$-SNE plots in the testing dataset, the average error is calculated over the $n$ $t$-SNE plots.

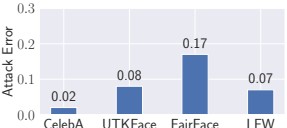

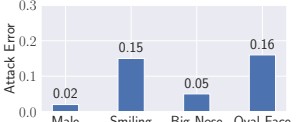

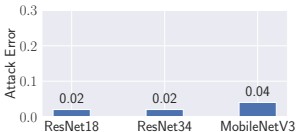

(a) Regression error on male proportion of different datasets.

(b) Regression error on proportion of different target properties of CelebA.

(c) Regression error on male proportion of CelebA with different models.

Figure 2: Regression results under setting 1 for different datasets, different target properties and different models. The testing data is from the same distribution as the training data. For (a) and (b), both the target model and shadow model are pre-trained ResNet18.

**Different Datasets.** We first conduct experiments with the target and shadow models being identical. Figure 2(a) reveals the effectiveness of the attack on the male proportion inference on different datasets. The highest average error is 0.17, on FairFace, while the average errors on the other three datasets are all lower than 0.10. When inferring the proportion of males on CelebA, the attack model reaches the lowest error of 0.02. Note that the average error of random guessing is 0.34. These results are consistent with those in Figure 1(a), where the target property from FairFace is also difficult to infer. The results show that, after learning from the training $t$-SNE plots with sufficient

different properties (from 0.00 to 1.00 in our experiments), the attack model can precisely infer the continuous proportion.

**Different Target Properties.** Similar to the multi-classification tasks, the attribute affects the performance of the attack in regression tasks. From Figure 2(b), the inference performance on the proportion of Oval Face has the highest error of 0.16. The inference accuracy on Oval Face also lags behind in the multi-classification task.

**Different Models.** Figure 2(c) shows that the attack is effective when the model are ResNet34 or MobileNetV3. For example, when the target model and shadow model are MobileNetV3, the inference error on the male proportion of CelebA is 0.04. Although this is still slightly worse than ResNet18 and ResNet34, it is much more precise than the multi-classification case.

**Transfer Study.** Table 2 shows the results of transfer study on regression tasks under setting 1. Consistent with the results of multi-classification (see Table 1), the attack performance on an unseen dataset is poor, with the average errors ranging from 0.21 to 0.41, most of which are worse than random guessing. However, adding a small number of $t$-SNEs from the new dataset can boost the performance of our attack significantly (see Section 5.5 for details).

**Different Shadow Model.** When the target model (ResNet34 or MobileNetV3) is different from the shadow model (ResNet18), our attack still works well (see Table 4). For ResNet34, the inference error on male proportion of CelebA marginally increases compared to setting 1, from 0.02 to 0.03, while the error increases to 0.13 for MobileNetV3, but still much better than random guessing.

In summary, the results of regression tasks are in accord with those of multi-classification tasks. The attack performs well on the testing data that are from the same distribution as the training data, while performs worse on unseen datasets. However, it is still valid across different properties and target models. The precision of regression tasks further demonstrates the potential severity of information leakage from $t$-SNE plots is underestimated.

### 5.3 ABLATION STUDY: DIFFERENT $t$-SNE SETTINGS

In this section, we investigate whether our attack is effective with: (1) fewer training $t$-SNE plots for each label, and (2) fewer sample points in each $t$-SNE. All the experiments are conducted on CelebA, and the target property is the proportion of males. For all the experiments, the shadow model is pre-trained ResNet18. The target model can be ResNet18 (setting 1) or ResNet34 (setting 2), denoted by the legend in the figures.

**Number of $t$-SNE Plots for Each Class.** Figure 7 shows the performance of multi-classification and regression tasks. Generally, with more $t$-SNE plots for each class, the accuracy of multi-classification increases and the error of regression decreases, for both setting 1 and setting 2. For the multi-classification case, 50 $t$-SNE plots for each class (250 plots in total) can achieve an accuracy of almost 0.80, even if the target model is not the same as the shadow model. And the regression error is lower than 0.10 with only 10 $t$-SNE plots for each class (1,010 plots in total). This further demonstrates the vulnerability of $t$-SNE plots against the property inference attack.

**Density Settings.** Density denotes the number of sample points (images) represented in a $t$-SNE plot, which may change the geometric features of $t$-SNE plots and subsequently affects the performance (see Figure 10). We additionally generate $t$-SNE plots, each with 500 and 750 sample points from CelebA (CelebA_500 and CelebA_750 for short), for multi-classification tasks and regression tasks. The number of training plots and testing plots is the same as the default setting. Figure 8 and Figure 9 exhibit the inference performance of attack models trained on different density settings. The inference performance is the best when the attack model is trained on the plots with the same density setting as the testing plots. However, when the training set is CelebA_750, the inference performance is good on both CelebA_500 and CelebA_1000, under both setting 1 and setting 2. For example, in Figure 9(a) and Figure 9(c), the regression error trained on CelebA_750 is less than 0.04 and less than 0.15 respectively, for both of the lines in each figure.

### 5.4 POSSIBLE DEFENSES AND ADAPTIVE ATTACKS

To mitigate the attacks, we investigate defense approaches based on *disturbance*, including noising, rounding, and thresholding. The experiments are conducted over regression tasks on CelebA.

Table 5: Inference errors with different defense strategies. The successful defenses are in bold.

| Defense Methods | Inference Error | Defense Methods | Inference Error |
|---|---|---|---|
| No defense | 0.02 | E-R INT | 0.03 |
| **E-N 2*STD** | 0.48 | C-R 0.1 | 0.02 |
| **E-N 1*STD** | 0.30 | C-R INT | 0.05 |
| **C-N 5%*STD** | 0.48 | E-T 50% | 0.03 |
| **C-N 2%*STD** | 0.48 | E-T 75% | 0.02 |
| E-R 0.1 | 0.02 | | |

Table 6: Adaptive attack performance against the successful defense methods.

| Defense Methods | Adaptive Err. | Non-adaptive Err. |
|---|---|---|
| E-N 2*STD | 0.15 | 0.48 |
| E-N 1*STD | 0.04 | 0.30 |
| C-N 5%*STD | 0.25 | 0.48 |
| C-N 2%*STD | 0.24 | 0.48 |

Both the target model and shadow model are pre-trained ResNet18 and the target property is the proportion of males. Note that we consider this setting as the most challenging one for defenses as it achieves the highest attack performance in previous experiments.

**Gaussian Noise in Embeddings (E-N) and Coordinates (C-N).** We first test the defense of adding Gaussian noise to embeddings and the $t$-SNE coordinates. For the E-N, we add the Gaussian noise to each image embedding before fitting the $t$-SNE. The mean of the noise is set to 0, and the standard deviation is set to a percentage of the current embedding values' standard deviation. For the C-N, the Gaussian noise is added to $t$-SNE coordinates. Concretely, in the current $t$-SNE plot, the standard deviation of the noise added to x coordinates is set to a percentage of all x coordinates' standard deviation. The noise is added to y coordinates in the same way. Results in Table 5 show the effectiveness of such defense methods. With Gaussian noise of 1 and 2 times the original standard deviation added to the embedding values (E-N), the inference error increases by 0.28 and 0.46 respectively, compared to the no defense (0.02). With Gaussian noise of 2% and 5% original standard deviation added to the coordinates (C-N), the inference error increases by 0.46. To evaluate the effect on the visual utility of such defense methods, we draw example $t$-SNE plots in Figure 11. Visually, $t$-SNE protected by E-N-1 (Figure 11(b)) still shows two clear clusters, while E-N-2 (Figure 11(c)) severely jeopardizes the visual utility. In C-N-2% (Figure 11(d)) and C-N-5% (Figure 11(e)), sample points of two colors are slightly blended, suggesting that they also influences the visual utility of $t$-SNEs.

**Rounding Embedding Values (E-R) and Coordinates (C-R).** Rounding is to round embedding values or $t$-SNE coordinates to a specified number of decimal places, reducing the precision of the embedding values and $t$-SNE plots. We observe that when we round the values to 0.1 or integer, neither E-R nor C-R can defend the $t$-SNE plots from the attack, with the inference error almost unchanged.

**Threshold Embedding Values (E-T).** For this defense, in each embedding, only the largest $k$ percentage of values are retained, while other values are set to 0. From Table 5, the defense both fails when $k$ is set to 50 and 75, indicating that it is not effective to simply drop some values in the embedding. We suspect that the important information remains in the largest 50% values and can still be captured by the $t$-SNE.

**Adaptive Attack.** We then consider an adaptive adversary who is aware of the defenses and can add those defended $t$-SNE plots into the training datasets. Specifically, the training $t$-SNE plots are processed in the same way as the effective defenses, i.e. the training $t$-SNE plots are generated with noised embeddings, or noise is added to the coordinates in the plots, which are the two effective defenses. The results are exhibited in Table 6. With the adaptive attack, all the attack errors drastically decrease. For instance, the E-N 1*STD adaptive attack achieves a low error of 0.04, while the error non-adaptive attack on the defended $t$-SNE plots is 0.30. The error of the adaptive attacks on C-N 5%*STD is 0.25 and on C-N 2%*STD is 0.24, which both maintain relatively high. This result demonstrates the effectiveness of the C-N defense method. However, as we observe in Figure 11, C-N decreases the visual utility of $t$-SNE, which calls for more effective and harmless defenses.

## 5.5 OPEN-WORLD SETTINGS

In this section, we investigate the robustness of our attack under more realistic settings. The adversary can train the attack model with mixed datasets and infer properties of $t$-SNE plots generated by fine-tuned models.

**Mixed Datasets.** Previously, we observe that the attack model trained on the data with only 5 $t$-SNE plots for each class does not have good inference performance (see Figure 7(b)). Now for each proportion in the training dataset, we mix the 40 $t$-SNE plots generated from UTKFace with only

a small number of CelebA $t$-SNE plots, to evaluate our attack. For both setting 1 and setting 2, the shadow model is pre-trained ResNet18. For setting 2, the target model is pre-trained ResNet34. From Figure 12, by mixing only 5 CelebA plots for each class, the inference error on the testing CelebA decreases significantly, from more than 0.30 to 0.05, under both target model settings. This performance is almost identical to the performance of the attack model only trained on CelebA (0.02 and 0.03, see Figure 2(a) and Table 4). However, the performance on the UTKFace dataset does not improve, by adding CelebA plots to training data. These results suggest that some datasets (e.g. CelebA) may be more susceptible to such attack and can be accurately inferred even if the training $t$-SNE plots are mostly from another distribution.

**Attack on $t$-SNE Plots Generated by Fine-tuned Models.** In practice, the model owner may fine-tune a pre-trained model on a specific target task, e.g. classifying the gender. Then, they may consider generating $t$-SNE plots colored by the task label (e.g. the gender or the race, see Figure 13(a) and Figure 13(b)) using the embeddings extracted by the fine-tuned model to showcase the performance of the model. Besides the information that we assume the adversary to have in Section 3, the adversary also has knowledge of the target task. The adversary then fine-tunes another pre-trained model on the same task, using the shadow dataset. We conduct the attack with regression tasks in this scenario. The target model is pre-trained ResNet18 fine-tuned on the target task. We randomly sample 10,000 images from the target (shadow) dataset to fine-tune the target (shadow) model, 8,000 for training and 2,000 for testing. Note that we here train the shadow model and target model under the same hyperparameters, with training epochs 8, batch size 64, learning rate 1e-4, and Adam the optimizer. In Table 7, the fine-tune task accuracy is shown in the Target Acc. and Shadow Acc. column.

From the results in Table 7, we observe that the inference performance is good even when the shadow model (ResNet18) is different than the target model. The inference error for ResNet34 is 0.10 while for MobileNetV3 is 0.13. These results reveal that our attack remains valid when the shadow model does not share parameters with the target model, as they are fine-tuned on non-overlapping data, which demonstrates that the threat to data privacy is underestimated in this practical scenario.

## 6    LIMITATIONS

**Limited Scenarios.** Note that our attack infers the property of the data that are in the $t$-SNE plot. One of the possible scenarios of our attack is to audit the ML model fairness. For instance, a model owner may claim the "good" performance of its model by a $t$-SNE plot with a specific property, e.g. 100% light-skinned people, as the "good" performance may be biased toward them (Buolamwini & Gebru, 2018). Our attack can uncover such unfairness by inferring the property of the $t$-SNE. However, if the data owner publishes a $t$-SNE plot with a property that is not presented in the original dataset (e.g. the proportion of males is not the same), without any performance bias, the attack may be not be useful. This is one limitation of the attack.

**Embedding Layer Specification.** In this paper, we assume that it is a common practice to use the outputs of the second-to-the-last layer as image emebeddings. For instance, Vision Transformer (ViT) (Dosovitskiy et al., 2021) uses the output of the classification token (the second to the last layer) to feed in the linear classifier (the last layer) for classification. However, the outputs from other layers can also be used for the $t$-SNE generation, which may render our attack invalid, as the attacker is unware of which layer to use. This is a limitation of our work and we leave the exploration for the future.

## 7    CONCLUSION

In this paper, we perform the first property inference attack through $t$-SNE plots. Experiments show that our attack is effective in inferring undisclosed properties of data in a $t$-SNE plot, i.e. the distribution of a certain attribute in the $t$-SNE plots, and even precisely with regression models. To mitigate the attack, we evaluate several defense mechanisms. Our evaluation result shows that adding noise to the embeddings or the $t$-SNE coordinates can defend against the attack. However, the defense of adding noise to coordinates can affect the utility of the $t$-SNE plots or can be nullified if the adversary adaptively conducts the attack. In the future, we plan to investigate the vulnerability of different attributes and datasets, as well as the effective transfer attack and defense methods.

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

## A APPENDIX

### A.1 DATASETS

**CelebA.** CelebA (Liu et al., 2015) is a large-scale face attributes dataset. There are more than 202K face images, each with 40 attribute annotations. We focus on the following attributes of CelebA: Male, Smiling, Big Nose, Oval Face. These attributes not only include some whole facial features (Gender, Oval Face), but also local detailed features (Big Nose, Smiling). The attack model will infer the proportion of each attribute of the samples used to generate the $t$-SNEs.

**LFW.** LFW (Huang et al., 2008) were originally provided for face verification, also known as pair matching under both restricted and unrestricted settings. The 13K images were further labeled with 73 attributes by attribute classifiers from Kumar et al. (2009). Many of the attributes overlap with CelebA. We use the deep-funneled version (Huang et al., 2012) of the data to infer the proportion of the same attributes as CelebA for the ablation study.

**UTKFace.** UTKFace dataset (Zhang et al., 2017) contains 23,708 face images with a long age span from 0 to 116 years old. Each image is labeled with age, gender, and ethnicity. In our experiments, we will infer the proportion of males and white people.

**FairFace.** FairFace dataset (Kärkkäinen & Joo, 2021) was constructed to mitigate the race bias problem in common datasets. A total of 108,501 images labeled with gender, race and age are collected from the YFCC-100M Flickr dataset and balanced on 7 race groups, i.e. White, Black, Indian, East Asian, Southeast Asian, Middle Eastern, and Latino. The images were split into training set and validation set in the original work. In this paper, we use the training set with 86K images. For the ablation study purpose, we will infer the proportion of males and white people as UTKFace.

### A.2 MODELS

**ResNet.** ResNet (He et al., 2016) is a deep learning model family, which utilizes residual learning to ease the training of deep networks. In this paper, we use the pre-trained ResNet18 and ResNet34 as the target model and the shadow model.

**DenseNet.** DenseNet (Huang et al., 2017) adds connections between each layer and every other layer. It mitigates the vanishing-gradient problem and reuses features between layers. We use the pre-trained DenseNet121 as the attack model for all the experiments to infer the dataset property of $t$-SNEs.

**MobileNet.** MobileNet (Howard et al., 2017) uses depthwise separable convolutions to reduce the calculation during convolution. In this paper, we use MobileNetV3_small (Howard et al., 2019) as the target model and the shadow model.

### A.3 GRAD-CAM VISUALIZATION

We try to apply Grad-CAM heatmaps (Selvaraju et al., 2017) to $t$-SNE plots of different proportions of males from CelebA to show the characteristics that the attack model pays attention to. The attack model is trained on $t$-SNE plots generated by ResNet18 for multi-classification, and we randomly select three plots for each proportion (0.3, 0.4, 0.5, 0.6, 0.7) from the CelebA test data, which are also generated by ResNet18.

In Figure 3, the first row presents the original plots for the second row, which are hard to distinguish by human eyes. However, from the heatmaps, we notice that the most concentrated part of plots with label 0.3 is the pivotal characteristic, while for 0.4, the attack model attends to detect a more diffuse part. For 0.5, the model focuses on the most sparsest part, and for 0.7, the focal characterstic is the sparse points between two groups of dense points. These different patterns are distinguishable for the attack model, but not that visually clear.

### A.4 ADDITIONAL RESULTS

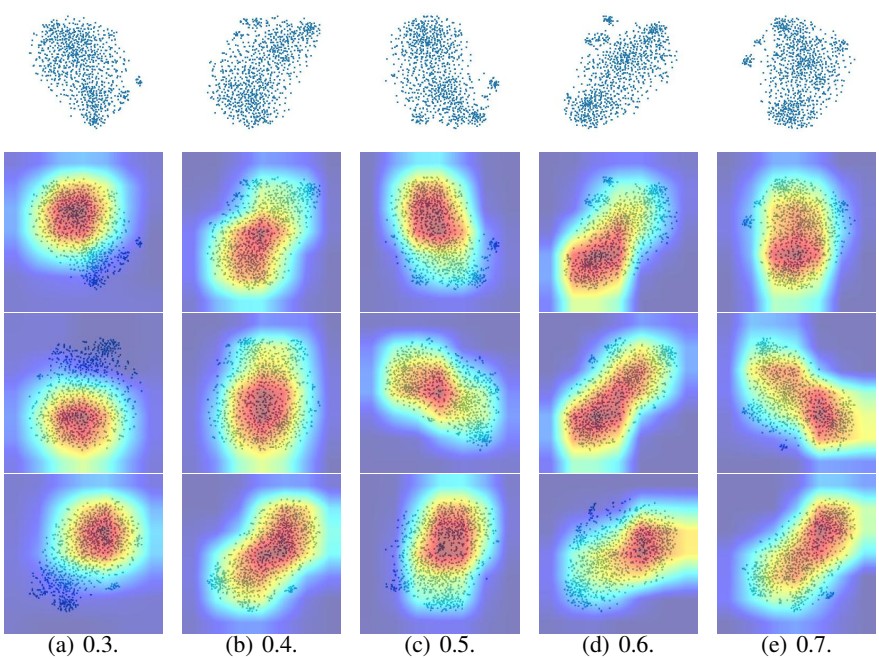

(a) 0.3.          (b) 0.4.          (c) 0.5.          (d) 0.6.          (e) 0.7.

Figure 3: Grad-CAM heatmaps of $t$-SNE plots of different proportions of males from CelebA. There are 1000 data points in each plot. The first row are the original plots of the second row.

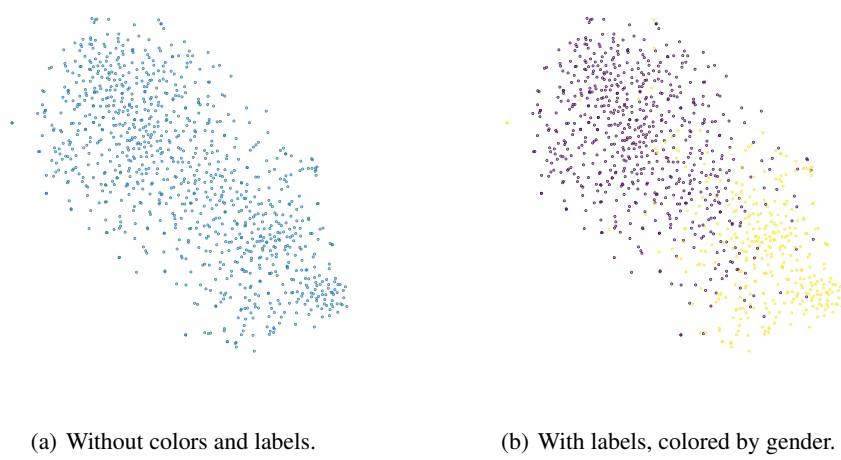

(a) Without colors and labels.                    (b) With labels, colored by gender.

Figure 4: $t$-SNE plots of the same proportion (0.0) of Big Nose faces from CelebA. The images are embedded by pre-trained ResNet18.

Table 7: Attack in the fine-tune scenario. The target (shadow) accuracy is the accuracy of the target (shadow) model on the target task. We then use the fine-tuned target (shadow) model to generate testing (training) $t$-SNE plots. The shadow model is fine-tuned ResNet18.

| Target Task | Target Model | Target Acc. | Shadow Acc. | Target Property | Test Err. |
|---|---|---|---|---|---|
| CelebA Male Classification (binary) | ResNet18 | 0.98 | 0.97 | Smiling | 0.08 |
| CelebA Male Classification (binary) | ResNet34 | 0.96 | 0.97 | Smiling | 0.10 |
| CelebA Male Classification (binary) | MobileNetV3 | 0.96 | 0.97 | Smiling | 0.13 |

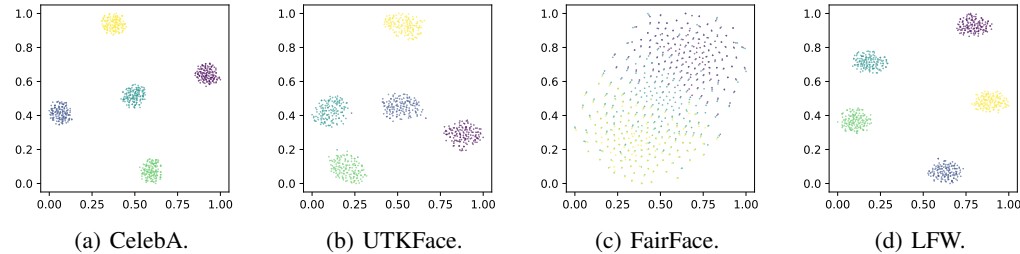

Figure 5: $t$-SNE plots of average embeddings of $t$-SNEs from different datasets, clustered by the proportion of males in the $t$-SNEs, scaled to $[0, 1]$. The average embedding of a $t$-SNE is the average embedding of all the image embeddings in this $t$-SNE. The image embeddings here are generated by pre-trained ResNet18.

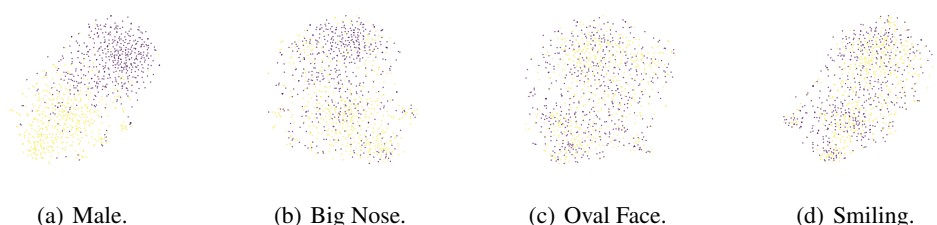

Figure 6: $t$-SNE plots of different attributes, with the same proportion (0.5) of faces with this attribute, sampled from CelebA, colored by the attribute. The feature extractor is pre-trained ResNet18.

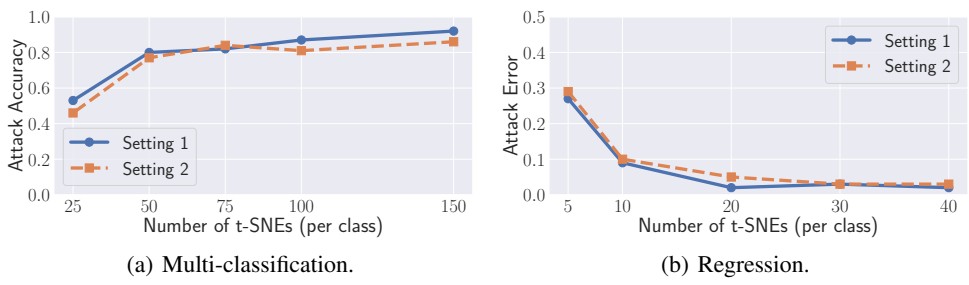

Figure 7: Inference Performance by using different number of $t$-SNE plots.

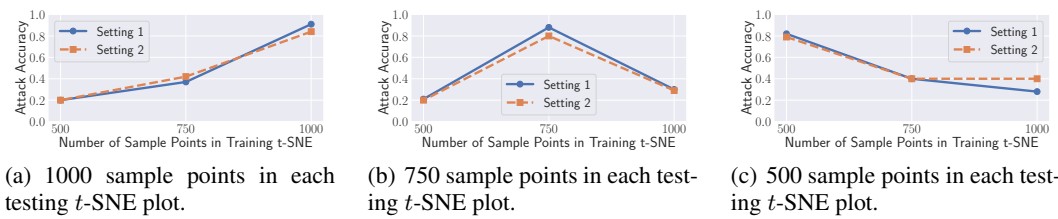

Figure 8: Multi-classification Inference Performance at different density settings on CelebA.

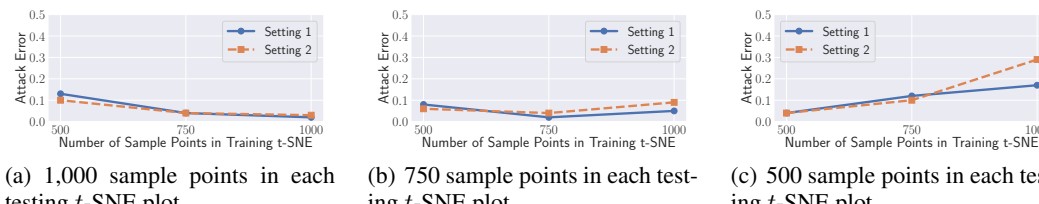

(a) 1,000 sample points in each testing $t$-SNE plot.

(b) 750 sample points in each testing $t$-SNE plot.

(c) 500 sample points in each testing $t$-SNE plot.

Figure 9: Regression inference performance at different density settings on CelebA.

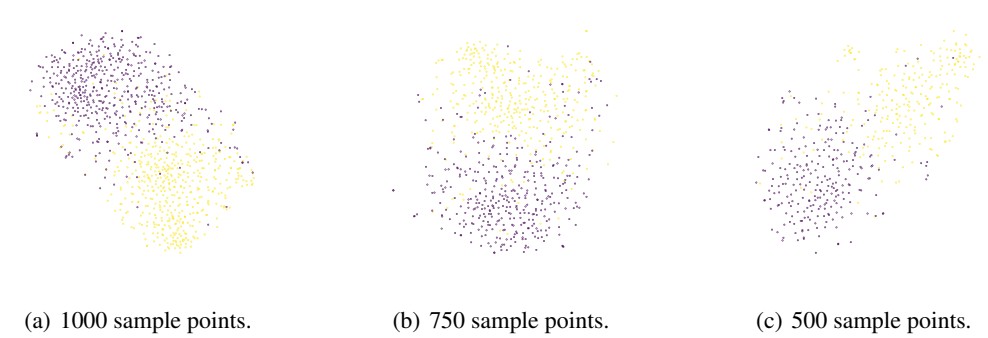

(a) 1000 sample points.

(b) 750 sample points.

(c) 500 sample points.

Figure 10: $t$-SNEs of face images (50% males) from CelebA with different density settings, colored by gender.

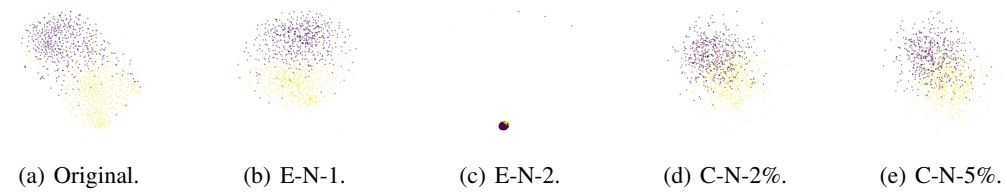

(a) Original.

(b) E-N-1.

(c) E-N-2.

(d) C-N-2%.

(e) C-N-5%.

Figure 11: $t$-SNEs of face images (50% males) from CelebA, colored by gender, perturbed by different defense methods.

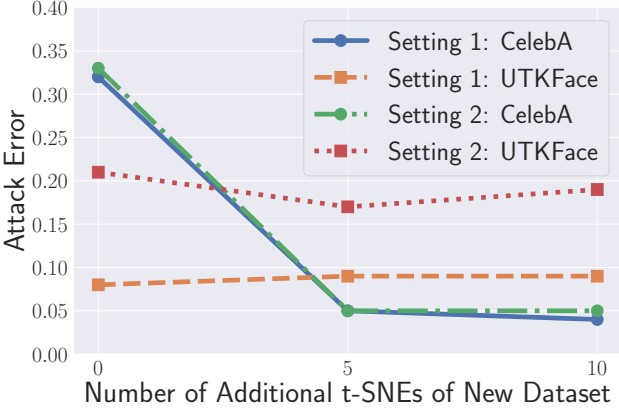

Figure 12: Regression inference performance on male proportion with mixed training $t$-SNE plots.

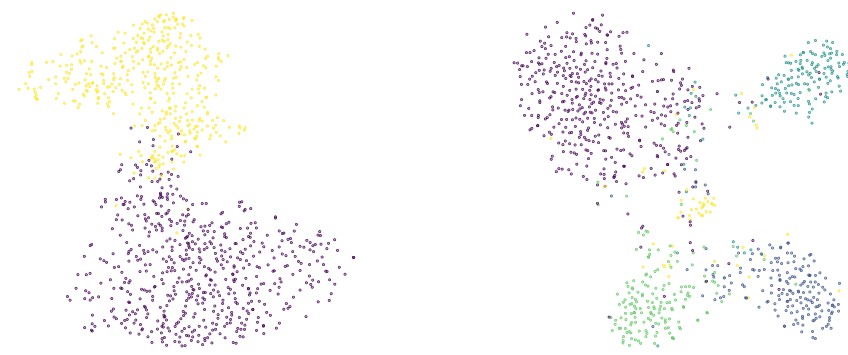

(a) Fine-tuned on CelebA Male classification task. Sample points are from CelebA, colored by gender, 50% of which are Smiling faces.

(b) Fine-tuned on UTKFace Race classification task, colored by the race. Sample points are from UTKFace, colored by the race, 50% of which are Male faces.

Figure 13: $t$-SNE plots generated from fine-tuned models. There are 1000 sample points in each $t$-SNE plot.

