# OpenReview forum: "Property Inference Attacks Against t-SNE Plots"
_ICLR.cc/2023/Conference — Submitted to ICLR 2023_

### Official Review · Reviewer_sG4d · 2022-10-19

**Confidence:** 5
**Correctness:** 3
**Technical Novelty And Significance:** 2
**Empirical Novelty And Significance:** 1
**Recommendation:** 3

**Clarity, Quality, Novelty And Reproducibility:**

__Clarity__: The paper is well written and presented for the most part (see minor comments below), and is easy to follow even without a substantial background in property inference.

__Quality and Novelty__: While the problem itself is very unique and original, I am not convinced of the threat posed in the paper. The proposed techniques are standard to image classification and property inference- a CNN (serving as a meta-classifier here) trained for property prediction.

__Reproducibility__: The main attack idea is straightforward, and sufficient details are provided in the paper (and the Appendix) to be able to reproduce results in the paper.

### Minor comments
- Section 3.1: "Such plots are commonly published on...". Can the authors please include some examples or references for this? Personally, I have never seen t-SNE plots for data outside whitepapers.

- Page 7, 'Different Target Properties' just like a baseline for accuracy for random guessing, the authors should also include a baseline for the case of regression. A straight-forward setting would be one where the adversary always predicts a constant value (0.5), or randomly picks a value in [0, 1].

**Strength And Weaknesses:**

## Strengths

- This work introduces a new kind of threat model- information leaked by visualizations. It is common to use techniques like t-SNE or PCA to visualize high-dimensional data; often to demonstrate the effectiveness of an approach in learning useful representations. This work tackles the threat posed by the release of such plots- specifically, properties related to the data used to generate these plots.
- Evaluations across different pairs of datasets are useful in better assessing the practicality of this risk; fine-tuning-based baselines are also useful in evaluating just how much data is actually needed by the adversary to launch successful attacks. This also holds true for evaluations with different class-wise plots and density settings.
- The inclusion of adaptive attacks and evaluations with mixed datasets (Section 5.5) is also useful and further reinforces that the information leaked in this scenario is fairly robust across several configurations.

## Weaknesses
- The property inference attack here really focuses on the data used to generate the plot, not the actual training dataset/distribution itself. A victim is unlikely to use all of its data for t-SNE visualization (given its computational cost). A straightforward defense would thus be to sample the t-SNE plot data from the test set such that the "property" is different from what was used for training. In this case, even if an adversary can perfectly predict the property of the plot data, the information gleaned has utility no longer. Even if the adversary does learn this information, it is not very useful- definitely not for model auditing. The victim can always claim that the data used for plot generation was differently distributed from the data used for training (which is really what auditors would care about).

- Section 3.2: "...and get the image embeddings". How is the adversary aware of which layer's features were used for the embeddings? How robust is the adversary to settings where this information is not known, i.e. the victim and adversary use different layer outputs for the plots?

- Section 2: "....most assume that the adversary has white-box access to the model." This is not true- there are several works [1, 2, 4] in the literature that propose and evaluate black-box attacks. Similarly, the claim on "most of the works infer binary properties" is not true either. Several works have studied various kinds of graph-related properties [3], as well as direct regression of these values [2].

- As demonstrated in Tables 1 and 2, performance is near-random (and even worse) when the training and testing data are different. Although this can be alleviated by using some data from the target distribution (as demonstrated later in the paper), this shows how sensitive the attack is to know the exact data distribution. This might not be feasible for parties that might actually release these plots (like a company talking about their new ML model), and it is even more unclear if this attack would be practical in such a scenario.

- Can plot rotation/scaling be a potential defense? It may not be perfect, but the adversary might suffer performance losses if trying to incorporate these invariances in its classifiers. Scaling the plot image, or rotating it, should not impact the utility of the t-SNE plot visualization and thus a victim could easily perform this post-processing to make things harder for the adversary.

### References
[1] Zhang, Wanrong, Shruti Tople, and Olga Ohrimenko. "Leakage of Dataset Properties in {Multi-Party} Machine Learning." USENIX Security Symposium. 2021.

[2] Suri, Anshuman, and David Evans. "Formalizing and Estimating Distribution Inference Risks." Privacy Enhancing Technologies Symposium. 2022.

[3] Zhang, Zhikun, et al. "Inference Attacks Against Graph Neural Networks." USENIX Security Symposium. 2022.

[4] Juarez, Marc, Samuel Yeom, and Matt Fredrikson. "Black-Box Audits for Group Distribution Shifts." arXiv preprint arXiv:2209.03620 (2022).

**Summary Of The Paper:**

This work proposes property inference attacks using t-SNE plots of data. While most work in the literature focuses on white-box or black-box model access, this threat model lies somewhere in the middle- predicting properties of hold-out data using 2-dimensional representations graphed onto images. The approach consists of data cleaning and using plots directly as input to a meta-classifier, which is trained by the adversary for property prediction. The authors demonstrate the robustness of their attack across different plot settings and propose a simple adaptation of their attack that can bypass potential defenses like noise addition in the latent space.

**Summary Of The Review:**

This work introduces a very new kind of threat model- one where the adversary tries to gain information about confidential data, given t-SNE plots generated with a model. The authors demonstrate how this leakage is very real, and fairly robust across various datasets, and configurations, and can be adapted to defenses with simple modifications. However, the practicality of the threat model is uncertain. The proposed attack cannot be used for model auditing (plausible deniability by the model trainer, as training data is different from plot-generation data), and it is unclear how the information leaked by the plots is of any use to the adversary. Even in the case that it is, the victim can always sample selectively from its test data, such that the leaked information is in no way relevant to the training distribution- like a 'honeypot' for the adversary.

---

> ### Author Response · Authors · 2022-11-15
> **Response to Reviewer sG4d**
>
> We would like to thank the reviewer for their insightful comments and feedback. Below are our responses to the comments and modifications made to the submission
>
> “The property inference…”
> - The data owner can publish t-SNE plots with data from a distribution different from the original dataset. However, the goal of publishing t-SNE plots is to directly reflect the dataset characteristics or model performance and thus the data points used for plotting should be randomly sampled from the distribution. Deliberately selecting data points with different attribute or property distributions will jeopardize the validity of the t-SNE plots.
>
> “Section 3.2…”
> - We admit that the victim may use another layer’s output as the image embedding. However, it is a common practice to use the second-to-the-last layer as the feature embeddings which contains the most useful information. For example, ViT (Vision Transformer) [1] uses the output of the classification token (the second to the last layer) to feed in the linear classifier (the last layer) for classification.
>
> “Section 2…”
> - We agree with the reviewer and will modify our writings accordingly.
>
>  “As demonstrated…”
> - We agree with the reviewer that our attack performs poorly on transfer tasks. Our paper is the first work revealing property information leakage from t-SNE plots and aims to draw attention to this vulnerability, considering our straightforward attack methodology already shows success in many settings. As far as we know, there are some learning methods that can improve the transfer performance, for instance, test-time training [2, 3]. With these methods, the attacker may be able to infer the properties of different datasets. These can be our future research directions.
>
> “Can plot…”
> - In fact, when training the attack model, we already randomly rotated the t-SNE plots as data augmentation. Scaling is also a trivial image modification, and the attacker can easily scale it back to the original scale using a photo editing software.
>
> Minor 1:
> - The author of t-SNE method[4] lists several examples on his personal website,  https://lvdmaaten.github.io/tsne/.
>
> Minor 2:
> - The baseline for regression tasks is 0.3366, which we mention in Section 5.2 “Different Datasets”. It can be proved mathematically or can be inferred from computer simulation. The python code for simulation is:
> import random
> errors = 0
> for i in range(1000000):
>     ground_truth = random.randint(0, 100) # randomly pick a ground truth proportion
>     guess = random.randint(0, 100) # randomly guess one
>     errors += abs(ground_truth - guess) # add to the error
> errors /= 10000000
> print(errors) # result: 33.6587515
>
> References:
> [1] Dosovitskiy, Alexey, et al. "An image is worth 16x16 words: Transformers for image recognition at scale." arXiv preprint arXiv:2010.11929 (2020).
> [2] Sun, Yu, et al. "Test-time training with self-supervision for generalization under distribution shifts." International conference on machine learning. PMLR, 2020.
> [3] Wang, Dequan, et al. "Tent: Fully test-time adaptation by entropy minimization." arXiv preprint arXiv:2006.10726 (2020).
> [4] Van der Maaten, Laurens, and Geoffrey Hinton. "Visualizing data using t-SNE." Journal of machine learning research 9.11 (2008).

---

> > ### Comment · Reviewer_sG4d · 2022-11-15
> > **Thanks**
> >
> > Thanks for the clarifications.
> >
> > - "...Deliberately selecting data points with different attribute or property distributions will jeopardize the validity of the t-SNE plots."
> >
> > I disagree: the point of t-SNE visualizations is to show that a model has learned good feature representations for the given distribution, and should not drastically change (in terms of getting the point across) if the model trainer uses 50% males, 50% females v/s 25% males, 75% females. It may not longer be a true "sample at uniform" but it definitely doesn't jeopardize the validity of the plots. i think this is a valid counter-measure and should be included as part of limitations
> >
> > - "For example, ViT ..."
> >
> > Thanks for the clarification. Please add this explicitly in the limitations (the part about not knowing the layers), and the example on ViT in the introduction (to justify this assumption).
> >
> > - "..for instance, test-time training"
> >
> > The test-time training scenario makes assumptions that may be a bit too much. If the adversary has access to the exact distribution, it is realistic to assume it is based on uniform sampling. In this case, why not directly measure the ratio/property of this data?
> >
> > - "The author of t-SNE method..."
> >
> > These examples are related to datasets that are already available. My point was: if a trainer has private data, it is unlikely they would proceed with this step in the first place.

---

> > > ### Author Response · Authors · 2022-11-16
> > > **Response to Reviewer sG4d**
> > >
> > > Thank you for your reply.
> > > We add the limitations section in our revised version.
> > >
> > > “I disagrees: …”:
> > > - We agree that for the described cases, it is a limitation. However, there is evaluation bias [1] in many ML models, which means that the evaluation is biased toward some specific data. For instance, [2] evaluated 3 commercial gender classification systems and found that in the gender classification task, darker-skinned females are the misclassified with error rates of up to 34.7% and while for lighter-skinned males the error is 0.8%. In this case, the model owner may deliberately choose the lighter-skinned people to generate t-SNE plots to show the performance of the gender classification model to show the “good” performance. Our attack can serve as an auditor for this kind of unfairness.
> > >
> > > “Thanks for clarifications …”:
> > > - Thank you. We will add this to the limitations.
> > >
> > > “Why not directly infer?”:
> > > - The attacker can collect images from the same dataset, but the property distribution may not be the same as the target dataset. For instance, the attacker may have images with 50% male (surrogate dataset), but the target dataset contains 40% males, and both of them are from CelebA.
> > >
> > > “These examples are related to datasets that are already available.”:
> > > - We admit that if the dataset is totally private, our current attack will be ineffective. We hope to see future work developing more sophisticated attack methods that allow transfer across datasets.
> > >
> > >
> > >
> > > [1] Suresh, Harini, and John V. Guttag. "A framework for understanding unintended consequences of machine learning." arXiv preprint arXiv:1901.10002 2 (2019): 8.
> > >
> > > [2] Buolamwini, Joy, and Timnit Gebru. "Gender shades: Intersectional accuracy disparities in commercial gender classification." Conference on fairness, accountability and transparency. PMLR, 2018.

---

### Official Review · Reviewer_6zD4 · 2022-10-24

**Confidence:** 4
**Clarity, Quality, Novelty And Reproducibility:** see above
**Correctness:** 4
**Technical Novelty And Significance:** 2
**Empirical Novelty And Significance:** 2
**Recommendation:** 5

**Strength And Weaknesses:**

In my opinion, the main strength of the paper is to suggest the framework of "attacks against t-SNE plots", which I believe is a novel concept and has not really been considered by anybody before.

The main weakness is that the suggested attack is rather trivial; also, it can only work in very limited situations where the shape of the t-SNE plot is strongly influenced by the fraction of males (or other similar property) and when the attacker has access to the training ("shadow") labeled data and can generate multiple t-SNE plots with varying property values.

The paper also has some presentation issues but those can be easily fixed. Overall I am giving it a borderline reject score.

PRESENTATION ISSUES

* My understanding is that the "attack model" treats t-SNE plots as images. If this is correct, then I think it should be made more explicit and stated more clearly, e.g. in the end of Section 4, and also elsewhere. If t-SNE plots are images, then what is their size in pixels? What size (in pixels) are the dots of the scatter plot? Are these images black-and-white? Grayscale? Etc.

* The entire evaluation setup was not sufficiently clear to me from Section 4. E.g. what does 0.92 accuracy (beginning of Section 4.1) mean? This is the average over what exactly? Different random splits of the data? What is the true proportion of males in those splits? Were the splits generated such that the true proportion could be .3, .4, .5, .6, .7 with equal probability? Please clarify.

* I think the paper *NEEDS* a figure (preferably Figure 1), that would show a t-SNE of the CelebA dataset under attack, and example t-SNE plots generated from shadow data with .3, .4, .5, .6, .7 fractions. This figure would illustrate the attack and would show that it is indeed possible to "guess" the fraction of males in the CelebA dataset correctly, because the shadow t-SNE plots all look sufficiently different from each other. I was not able to find such figure either in the main text, or in the appendix.

**Summary Of The Paper:**

The paper "Property inference attacks against t-SNE plots" suggests a way to infer some properties of the data set from an unlabeled t-SNE plot of that data set. For example, if the data contain images of males and females, then the attack would aim to infer the fraction of males from the shape of the unlabeled t-SNE plot. To do this, the attacker generates multiple t-SNE plots of the data with varying fraction of males, and then trains a classifier to predict the fraction of males from the t-SNE plot (treated as image). This classifier can then be applied to the t-SNE plot under attack. The paper shows good performance on some of the datasets.

**Summary Of The Review:**

Interesting idea, but a rather trivial attack with very limited applicability. Borderline reject.

---

> ### Author Response · Authors · 2022-11-15
> **Response to Reviewer 6zD4**
>
> We would like to thank the reviewer for their insightful comments and feedback. Below are our responses to the comments and modifications made to the submission.
>
> “t-SNE plot is strongly influenced by…”
> - We admit that the attack is only effective when the shape of t-SNE plots is affected by the fraction of a certain attribute. This is why some attributes are easily attacked while others are not. However, our discovery that many properties in the dataset indeed can affect t-SNE plots is the key intuition behind the attack.
>
> “the attacker has access to the labelled data.”
> - It is a realistic assumption as many image datasets are collected from the Internet. Therefore the attacker can perform similar crawling. After obtaining the data, they can either manually label them or use ML models to predict the labels. The labels of the LFW dataset used in our experiments are predicted by the model from another paper (see LFW website for details, http://vis-www.cs.umass.edu/lfw/). With the labels, the attacker can sample any number of them to generate a t-SNE plot with a specific proportion of males (e.g. 50% are males. They can simply randomly choose 500 males and 500 females from the shadow dataset). In this way, the attacker can gain the training t-SNE plots for the attack model.
>
> Presentation issues:
> - We add clarification for the pixel size of t-SNE plots in  Section 4 “Experimental Setup” - “t-SNE Generation”. The pixel size is 300 * 300. The plot uses one color, the default python blue. Please see Figure 4(a) for an example.
> - The settings are presented in Section 4 “Experimental Setup” - “Target/Shadow Model and Two Experiment Settings” and “Experimental Setup” - “t-SNE Generation”. One setting is the target model is the same as the shadow model, while the other setting is the target model is different from the shadow model. In Section 5, all the experiments use Setting 1 except the “Different Shadow Model” part. For example, 0.92 mentioned in the review is under Setting 1, with both the target model and shadow model being ResNet18. We generate 150 t-SNE plots for each of .3, .4, .5, .6, .7 (proportion of males) using shadow models, on which the attack model is trained. And 50 plots for each are generated using target models, on which the attack model is tested. 0.92 is the accuracy for this test.
> - In fact, different t-SNEs are hard to distinguish by human eyes. We use Grad-CAM to identify the pattern differences of the plots. The results are added to the appendix, please check.

---

> > ### Comment · Reviewer_6zD4 · 2022-11-16
> > **Distinguishing CelebA t-SNEs**
> >
> > Thank you for your reply.
> >
> > > In fact, different t-SNEs are hard to distinguish by human eyes. We use Grad-CAM to identify the pattern differences of the plots. The results are added to the appendix, please check.
> >
> > To be honest, after seeing the new Figure 4 in the Appendix, I am a bit confused by how this whole thing works.
> >
> > 1. I am not sure I understand the difference between the 1st row in Fig 4 and the next three rows. My understanding is that for each proportion (e.g. 0.3), you repeatedly sample from CelebA 1000 images with this proportion of males, and perform t-SNE. The figure shows 4 resulting t-SNEs for each proportion. Is that correct? Do you mean that the 1st row is your "test data" and the next three rows is your "training data" for the classifier?
> >
> > 2. To me all these t-SNEs look very very similar, so I have a really hard time believing that a five-class classifier can achieve 0.92 accuracy here. 92% is really high, but it seems a human would perform at chance level, which in this case is 20%. Do you agree? To be honest, this makes me doubt the reported results.
> >
> > Grad-CAM visualization is IMHO not very helpful here, I would rather see this figure without the Grad-CAM heatmaps.
> >
> > And I still think that at least the 1st row of this figure should be in the main text.

---

> > > ### Author Response · Authors · 2022-11-16
> > > **Response to Reviewer 6zD4**
> > >
> > > Thank you for your reply.
> > >
> > > “I am not sure I understand the difference between the 1st row in Fig 4…”:
> > > - Sorry for the confusion. All the tsne plots are from the test data. The first row is the same as the second row, except the Grad-CAM mask. The second, third, and fourth rows are generated with different seeds. For example, the three plots with Grad-CAM masks in column “0.3” are all generated by sampling 1000 points from CelebA (with different sampling seeds), among which there are 300 males.
> > >
> > > “To me all these t-SNEs look very very similar…”:
> > > - We agree that it is hard to distinguish the difference by human eyes. However, as we can see from the Grad-CAM, the model actually discovers different patterns from the plots generated by different proportions of males from CelebA. For these plots, we checked the model output and we confirmed that these plots are correctly predicted. If it’s necessary, we can put the second row (i.e., the t-SNE plots with Grad-CAM mask) of Fig.4 into the main text to better illustrate how the attack model differentiate those plots.

---

> > > > ### Comment · Reviewer_6zD4 · 2022-11-16
> > > > **Thanks**
> > > >
> > > > I just want to emphasize, that given Figure 4, your results are IMHO **very counterintuitive**, and I think the burden is on you to convince a skeptical reader that they really do hold...
> > > >
> > > > PS. This was not clear to me from the original submission. It only became clear after you added Figure 4 (thanks for that!).

---

### Official Review · Reviewer_5Mjd · 2022-10-28

**Confidence:** 4
**Clarity, Quality, Novelty And Reproducibility:** The paper was very clear, and the wor…
**Correctness:** 4
**Technical Novelty And Significance:** 4
**Empirical Novelty And Significance:** 4
**Recommendation:** 6

**Strength And Weaknesses:**

### Pros:
1) The paper is very clearly presented.
2) The idea is interesting, and as far as I know, novel.
3) Experimentation is thorough.

### Cons/Comments:
1) The results are lukewarm in my eyes. Even in the comparatively easier setting of proportion classification, the proposed attack requires surrogate and target model pretrained on the same distribution, and several labeled tsne plots on which to train the attack model. And even then, the attack success on some datasets is fairly weak.
2) Am I correct that the target model and the surrogate model are both pretrained on the same data for all experiments? If so, why? It seems like a much more realistic setting is that the target model is trained on a distribution about which the attacker has little information.
3) Many of the experimental settings seem a bit contrived. The more realistic setting (transfer setting in 5.5), is a step in the right direction, but here the authors only limit the number of tsne plots available to the attacker. Another constraint I would have liked to have seen explored in this setting is tsne plots generated only with a small amount of data. In fact, throughout the paper, it seems like the authors assume that that data on which to generate tsne plots with labeled proportions, can be easily found. This seems like a very strong assumption.

Overall, I'm a bit torn. The paper is a very interesting idea, and novel, but the results are a bit lukewarm and in somewhat contrived settings.

**Summary Of The Paper:**

The authors propose a property inference attack against tsne plots wherein an attacker can train a classifier on tsne plots from surrogate models in order to estimate sensitive characteristics about datasets for which other tsne plots were generated.

**Summary Of The Review:**

I vote for acceptance because I think the idea is quite novel, and interesting. However, I would really like the authors to address my concerns/questions about the settings for the experiments. I would consider lowering my score if the questions are not sufficiently addressed.

---

> ### Author Response · Authors · 2022-11-15
> **Response to Reviewer 5Mjd**
>
> We would like to first thank the reviewer for their insightful comments and feedback. Below are our responses to comments and modifications made to the submission.
>
> “The results are…”
> - As far as we know, many datasets are collected from the Internet. For instance, UTKFace is collected from the Internet, as claimed by the official website (https://susanqq.github.io/UTKFace/). Therefore, the adversary can also collect similarly distributed data from the Internet to train the surrogate model. Note that in this paper, our main goal is to point out the threat of property inference attack against t-SNE plots and we empirically show that, in many cases, the adversary can already mount effective attacks. It is possible that for some datasets, specific properties are harder to be inferred. We leave it as our future work to develop more advanced attack methodologies.
>
> “Am I correct…”
> - The target and surrogate models are indeed pretrained on the same data. When using t-SNE plots to display characteristics of datasets, the model serves only as a feature extractor. Commonly people use the simple pre-trained model to extract feature embeddings, such as ResNet18 pretrained on ImageNet. For instance, [5] uses ResNet50 as the feature extractor to detect breast abnormality.
>
> “Many of the…”
> - We perform an experiment on t-SNEs generated with only a small set of data (i.e. the shadow dataset is small). Note that in our paper, we simply divide the image dataset into two halves. For CelebA, the shadow dataset contains 101300 images. Now we first randomly sample 2100 images with 1050 males and 1050 females from the 101300 images, as the minor shadow dataset. This is only approximately 2 times (2100) the number of images in each t-SNE plot (1000). We use the ResNet18 as the target model and shadow model (setting 1). We then generate t-SNE plots with these 2100 images, that is, sample 1000 from the 2100 images to draw t-SNE. We conduct the regression task and results show that the average inference error is only 0.04 on CelebA test plots. Due to the time limit, we did not conduct more experiments. We believe that with only a small number of images, the attack can still work and thus largely obviates the need for a large shadow dataset.
> - Regarding the proportions, since the sensitive property is labelled in the surrogate dataset, the adversary can manually select data points with different proportions to generate the t-SNE plots and train the attack model.
>
> [5] Yu, Xiang, et al. "ResNet-SCDA-50 for breast abnormality classification." IEEE/ACM transactions on computational biology and bioinformatics 18.1 (2020): 94-102.

---

### Decision · Program_Chairs · 2023-01-20

**Decision:**

Reject

**Justification For Why Not Higher Score:**

The attack was of relatively low efficacy and works best in restricted settings.

**Justification For Why Not Lower Score:**

N/A

**Metareview: Summary, Strengths And Weaknesses:**

This paper shows how to perform perform privacy attacks against t-SNE plots. The reviewers were positive on the identification of the attack vector. However, they were negative on the strength of the attack and the assumptions required to execute the attack/when it would be effective. There were also some concerns about certain figures in the paper, but unfortunately code was not available to investigate further.